# *O*-GlcNAc Modification and Its Role in Diabetic Retinopathy

**DOI:** 10.3390/metabo12080725

**Published:** 2022-08-05

**Authors:** Chengzhi Liu, Wenkang Dong, Jun Li, Ying Kong, Xiang Ren

**Affiliations:** 1The First Affiliated Hospital of Dalian Medical University, Dalian 116011, China; 2Department of Histology and Embryology, College of Basic Medicine, Dalian Medical University, Dalian 116044, China; 3Core Laboratory of Glycobiology and Glycoengineering, College of Basic Medicine, Dalian Medical University, Dalian 116044, China

**Keywords:** *O*-linked β-N-acetylglucosamine modification, diabetic retinopathy, hexosamine biosynthetic pathway, retinal microvascular lesions, neurodegeneration

## Abstract

Diabetic retinopathy (DR) is a leading complication in type 1 and type 2 diabetes and has emerged as a significant health problem. Currently, there are no effective therapeutic strategies owing to its inconspicuous early lesions and complex pathological mechanisms. Therefore, the mechanism of molecular pathogenesis requires further elucidation to identify potential targets that can aid in the prevention of DR. As a type of protein translational modification, *O*-linked β-N-acetylglucosamine (*O*-GlcNAc) modification is involved in many diseases, and increasing evidence suggests that dysregulated *O*-GlcNAc modification is associated with DR. The present review discusses *O*-GlcNAc modification and its molecular mechanisms involved in DR. *O*-GlcNAc modification might represent a novel alternative therapeutic target for DR in the future.

## 1. Introduction

Diabetic retinopathy (DR) is a common chronic complication of diabetes worldwide and the leading cause of visual impairment and blindness among adults aged 20–74 years [1]. The global prevalence of DR from 2015 to 2019 was 27.0% in patients with diabetes, owing to the dramatic increase in the incidence of diabetes. This trajectory is predicted to continue in the forthcoming decades. Inconspicuous lesions in the early stage of DR make it challenging to explore effective therapeutic strategies [2].

Currently, intraocular treatment strategies for diabetic eye diseases include laser photocoagulation [3], intravitreous injections of Antagonists of Vascular Endothelial Growth Factor (Anti-VEGF) [4], steroid agents [5], and vitreoretinal surgery [6]. However, each has its limitations and side-effects, such as visual impairment and discomfort, cataract, and glaucoma. Approximately 20–30% of patients who undergo vitreoretinal surgery experience substantial vision loss after surgery and are at a higher risk of ocular venous air embolism [7]. These therapies are designed to treat advanced disease conditions, have a detrimental effect on patients’ quality of life, and impose a financial burden [8]. Therefore, exploring novel clinical diagnostic modalities and treatments is essential in the early stages of DR.

Multiple factors can lead to the occurrence and development of DR, including increased polyol-pathway activity [9], protein kinase C activation [10], oxidative stress [11], and inflammatory response [12]. *O*-linked N-acetylglucosamine (*O*-GlcNAc) modification is a unique form of post-translational protein modification (PTM). Recently, studies on *O*-GlcNAc modification have gained traction in various diseases, such as Alzheimer’s disease [13], Parkinson’s [14], cancer [15], and diabetes mellitus [16]. *O*-GlcNAc modification reportedly displays a strong correlation with DR [17,18]. However, the effect of *O*-GlcNAc modification on DR has not yet been fully elucidated. In this review, we summarize *O*-GlcNAc modification and its role in DR.

## 2. *O*-GlcNAc Modification

Protein translational modifications include changes such as phosphorylation, acetylation, ubiquitination, and glycosylation. After the discovery of *O*-linked N-acetylglucosamine by Torres and Hart in 1984, more than 5000 *O*-GlcNAcylated human proteins were identified [19,20]. *O*-GlcNAc modification is a critical and inducible PTM that involves the synthesis and addition of a single *O*-GlcNAc moiety to the hydroxyl groups of serine and/or threonine residues of proteins. It modulates biological processes such as cell-cycle progression [21], translation, transcription [22], nutrient sensing [23], and stress responses [24]. *O*-GlcNAc modification is required for normal physiological activities in mammals and plays a prominent role in embryonic stem-cell viability and embryonic development [25]. Altered GlcNAc concentrations disrupt the *O*-GlcNAc pathway and influence diverse aspects of cellular physiology [26].

### 2.1. Synthesis of UDP-GlcNAc via Hexosamine Biosynthesis Pathway

UDP-N-acetylglucosamine (UDP-GlcNAc), which is synthesized via the hexosamine biosynthesis pathway (HBP), acts as a substrate for β-N-acetylglucosaminyltransferase (OGT) and forms the *O*-GlcNAc modification [27]. HBP, a relatively minor branch of glycolysis, utilizes approximately 2–3% of all cellular glucose [28]. Therefore, *O*-linked-N-acetylglucosaminylation (*O*-GlcNAcylation) functions as a vital regulator of glucose metabolism [29]. The first two steps of the hexosamine biosynthesis pathway involve the hexokinase-catalyzed phosphorylation of glucose to glucose-6-phosphate and the phospho-glucose isomerase-mediated transformation into fructose-6-phosphate, which are similar to the glycolysis pathway. The HBP pathway then diverges from the glycolysis pathway. A small amount of F-6P is converted into glucosamine-6-phosphate (GlcN-6P) catalyzed by glutamine: fructose-6-phosphate amidotransferase (GFAT), which is the rate-limiting step of HBP and is regulated by feedback mechanisms [30]. GFAT has two isoforms: GFAT1, which is abundantly expressed in the skeletal muscle and heart [31], and GFAT2, mainly expressed in the central nervous system, including retinal neurons [32]. Subsequently, through a series of complicated chemical reactions, UDP-GlcNAc is synthesized and subsequently added to the serine and/or threonine residues of proteins (Figure 1).

### 2.2. Regulation of O-GlcNAc Modification by OGT and OGA

As the substrate of *O*-GlcNAc modification, changes in UDP-GlcNAc levels have an impact on the *O*-GlcNAc modification of many proteins [33]. The reversible and cyclic addition and deletion of UDP-GlcNAc are performed by OGT and *O*-GlcNAc hydrolase (OGA), respectively [34,35].

OGT, which is responsible for adding GlcNAc to serine and threonine residues, is encoded by the OGT gene residing on Xq13 [36]. It is ubiquitously expressed and was shown to exist in all metazoans, including humans, *C. elegans*, and rats [37]. The open reading frames of OGT genes are highly conserved in all these organisms [27]. Furthermore, OGT activity in the brain is ten times higher than that in muscle, adipose tissue, heart, and liver [28]. The enzyme has two major functional domains: the N-terminal domain, which consists of several tetra tripeptide repeats (TPR), and the C-terminal domain, which shows glycosyltransferase activity and binds UDP-GlcNAc [38]. Based on the number of TPRs, OGT isoforms can be divided into three types: full-length OGT (ncOGT; 13.5 TPRs), mitochondrial isoform (mOGT; 9 TPRs), and short isoform (sOGT; 2.5 TPRs) [29]. Unlike the other two isoforms, which accumulate in the nucleus and cytoplasm, mOGT tends to be present in the inner mitochondrial membrane [39]. The TPR domain directs the polypeptide sequence to the catalytic domain by binding to asparagine and aspartic acid [40]. In the retina, OGT-positive cells are primarily located in the inner nuclear and plexiform layers, ganglion cell layer and, in a later stage, photoreceptor inner segment [41,42].

OGA is responsible for removing GlcNAc from serine and threonine residues and is mainly enriched in the cytoplasm [35]. The OGA gene, which is also highly conserved, is located on the long arm of chromosome 10 (10q24). OGA has two isoforms: nucleocytoplasmic OGA (OGA-L), located in the cytoplasm and nucleus, and short OGA (OGA-S), located in the endoplasmic reticulum and lipid droplets [43].

### 2.3. Characteristics of O-GlcNAc Modification Compared with Other PTMs

*O*-GlcNAc modification has many characteristics compared with classical N-linked glycosylation. N-linked glycosylation is a diverse process that adds canonical multimeric long-chain glycan structures to extracellular proteins. In contrast, *O*-GlcNAc addition to serine and/or threonine residues mainly occurs on nuclear and cytosolic proteins. This modification is reversible through the cyclic action of OGT and OGA.

Increasing evidence demonstrates that *O*-GlcNAcylation is more similar to phosphorylation than to classical glycosylation [44]; both are rapidly cycling post-translational modifications. Their amino-acid modification sites include serine and threonine, leading to extensive interactions between the *O*-GlcNAcylation and phosphorylation mechanisms. *O*-GlcNAcylation and phosphorylation are frequently mutually exclusive, referred to as the ‘‘Yin–Yang” model [45]. However, the specificity of phosphorylation sites increases with increased *O*-GlcNAcylation levels [46]. They are also susceptible to nutrients because their donor substrates are high-energy products of cellular metabolism.

## 3. Relationship between *O*-GlcNAc Modification and Diabetes

Diabetes, a metabolic disorder, is characterized by hyperglycemia, insulin secretion defects, and insulin insensitivity. Diabetic hyperglycemia is caused by defective biological effects of insulin (type 2 diabetes mellitus) resulting from obesity [47] or decreased insulin secretion (type 1 diabetes mellitus) [48]. OGT, a recognized cellular nutrient sensor of the systemic metabolic state, is abundant in the pancreas [49]. The deletion of the OGT gene in the pancreatic epithelium can lead to pancreatic hypoplasia [50]. *O*-GlcNAcylation regulates β-pancreatic-cell survival as well as insulin secretion and β-cell capacity under physiological conditions [51]. As mentioned in this review, OGT is highly expressed under high-glucose conditions. Furthermore, insulin can phosphorylate OGT via its receptor, contributing to increased OGT activity [52]. OGT overexpression is related to type 1 and type 2 diabetes mellitus.

### 3.1. Type 2 Diabetes Mellitus and O-GlcNAc Modification

Type 2 diabetes mellitus is characterized by impaired biological effects of insulin and is prevalent in more than 90% of the reported cases. Insulin resistance is a primary reason characterized by reduced insulin activity despite its excessive levels in the blood, eventually leading to sustained hyperglycemia. In other words, when the body is in a state of insulin resistance, higher insulin concentrations are required to activate its receptors for normal physiological functions owing to reduced insulin sensitivity. The hexosamine biosynthesis pathway and *O*-GlcNAc modifications may play crucial roles in this process (Figure 2). In 1991, Marshall et al. first suggested that excessive glucose flux via HBP contributes to insulin resistance [28].

Activated-insulin-receptor signal transduction has two main branches: the insulin receptor substrate (IRS)–phosphoinositide 3-kinase (PI3K)–protein kinase B (Akt/PKB) pathway and the IRS–growth factor receptor-bound protein 2 (Grb2)–salt overly sensitive (Sos)–Ras–mitogen-activated protein kinase (MAPK) pathway [53]. The Akt/PKB serine/threonine kinase is a significant intracellular second messenger that phosphorylates and activates basic downstream kinases, such as glycogen synthase kinase 3, mammalian target of rapamycin complex 1, ribosomal protein S6 kinase, and transcriptional regulators such as Forkhead box O (FoxO) family members, sterol regulatory element-binding protein, peroxisome proliferator-activated receptor γ coactivator 1, and GTPase-activating protein Akt substrate 160 kDa [54]. In addition, glucose transporter-4 (GLUT4) is predominantly located in the cytoplasm, and Akt phosphorylation translocates the transporter to the plasma membrane for insulin-stimulated glucose uptake [55].

HBP contributes to insulin resistance mainly through the increased *O*-GlcNAc modification of insulin-signaling-pathway intermediates, which can be reversed with GFAT inhibitor treatment [56]. Therefore, we hypothesize that this phenomenon might result from an increased *O*-GlcNAc modification of intermediates, including IRS, Akt, FoxO, and other molecules, attenuating the insulin signaling cascade. Under physiological conditions, the normal signaling activities of certain intracellular proteins in pancreatic β cells are dynamically regulated via phosphorylation and *O*-GlcNAcylation in response to extracellular signals, which could cause diabetes. In animal models of insulin resistance and type 2 diabetes, reduced glucose uptake was associated with decreased phosphorylation of IRS-1 tyrosine [57]. Seung et al. treated primary rat adipocytes with an OGA inhibitor, O-(2-acetamido-2deoxy-D-glucopyranosylidene) amino-N-phenylcarbamate (PUGNAc), to increase *O*-GlcNAc modification. They observed that the increased *O*-GlcNAc modification of IRS-1 and Akt2 reduced the insulin-stimulated phosphorylation of IRS-1 and Akt2, which induced insulin resistance characterized by prominently decreased GLUT4 translocation in adipocytes [58]. Kaleem et al. discovered that the site modifications existed in both IRS-1 and IRS-2 (Ser1101 in IRS-1 and Ser1149 in IRS-2) [59] and that the excessive *O*-GlcNAcylation of these sites led to insulin resistance. The *O*-GlcNAcylation of IRS-1/2 can reduce the interaction with its downstream molecule, phosphoinositide 3-kinase (PI3K), which may be a reason for the attenuation of the insulin signaling cascade [60].

Phosphatidylinositol (3,4,5)-trisphosphate (PIP3), one of the molecules in the insulin pathway, recruits Akt to initiate signaling cascades during the early stages. OGT can also be recruited to the plasma membrane through the phosphoinositide-binding domain of OGT, which may induce insulin resistance [61]. In addition, the interactions between OGT and PIP3 also promote the *O*-GlcNAcylation of nuclear proteins, such as pancreas-duodenum homeobox-1 (Pdx-1). The *O*-GlcNAcylation of the Pdx-1 protein is positively correlated with the increase in its DNA binding activity [62]. Glucose stimulates the expression of G protein-coupled receptor 40 (GPR40) in the pancreas by increasing the binding between Pdx-1 and A-box in the HR2 region of the GPR40 promoter [63]. GPR40 activation, which leads to insulin secretion, has become an attractive target for type 2 diabetes treatment [64]. Collectively, the *O*-GlcNAc modification might play a conducive role in alleviating insulin resistance.

### 3.2. Type 1 Diabetes Mellitus and O-GlcNAc Modification

Type 1 diabetes is a complex disease involving multiple factors, such as genetic susceptibility, immune dysfunction, and inflammation, and is characterized by the destruction of insulin-producing β cells. In this review, we found that persistent hyperglycemia induces glucose toxicity through *O*-GlcNAc modification.

Thioredoxin-interacting protein (TXNIP), one of the primary mediators of β-cell dysfunction, inhibits glucose uptake by reducing GLUT1 mRNA expression [65], promoting the activation of the NOD-like receptor protein-3 (NLRP3) inflammasome, and promoting programmed cell death [66]. This protein is persistently elevated in diabetes [67] and subjected to *O*-GlcNAcylation. The interaction between TXNIP and its binding partner NLRP3 was quantitatively analyzed using the immunoblotting technique. A plasmid encoding a bioluminescence-resonance-energy-transfer biosensor comprising the pro-IL-1β sequence was used to quantitatively analyze its cleavage. OGT expression induced by PUGNAc and high glucose concentration reportedly contribute to the interaction between TXNIP and NLRP3 proteins, promoting pro-IL1β-cleavage-mediated inflammasome activation [68].

*O*-GlcNAcylation also induces β-pancreatic-cell death by interfering with Akt phosphorylation. Kang et al. treated rat pancreatic cells with glucosamine and discovered an increased *O*-GlcNAcylation of Akt Ser473; consecutively, the phosphorylation of this site was decreased. This competitive inhibition between phosphorylation and *O*-GlcNAcylation is associated with the apoptosis of murine β-pancreatic cells, leading to insufficient insulin secretion [69]. This observation could be attributed to the overexpression of pro-apoptotic protein Bim induced by impaired Forkhead box protein O1 (FOXO1) inactivation [70].

## 4. Relationship between *O*-GlcNAc Modification and Diabetic Retinopathy

DR is characterized by progressive and irreversible damage. It can be divided into two phases based on the occurrence of neovascularization in microvascular lesions: early (non-proliferative diabetic retinopathy (NPDR)) and advanced stages (PDR). NPDR has no apparent symptoms, including the blood–retinal barrier (BRB) breakdown [71], vascular-endothelial-cell and pericyte apoptosis, macular edema, microaneurysm, vascular-basement-membrane thickening, and capillary occlusion. It can go undetected in patients owing to the asymptomatic characteristics in the early stage. Patients experience vision loss during the advanced stage of PDR, which includes neovascularization usually accompanied by vitreous hemorrhage, traction retinal detachment, iris neovascularization, and angular neovascularization with elevated intraocular pressure (neovascular glaucoma) [72].

Recent research confirmed that DR is not just a microvascular disease but a combination of neurovascular diseases. In addition, retinal neurodegenerative changes in DR may emerge earlier than microvascular lesions, characterized by reactive gliosis and damage to photoreceptors [73,74]. Retinal angiopathy occurs in retinal arterioles in the initial stage, which further leads to the increase in retinal microvascular pressure, leading to a series of microvascular lesions. Previous studies focused on the microvascular lesions of DR. Since retinal neurodegenerative changes occur earlier, targeting retinal neurodegeneration is more conducive to the early detection and treatment of DR. We summarize that increased *O*-GlcNAc modification caused by diabetes-induced hyperglycemia involves microvascular lesions and neurodegeneration.

### 4.1. O-GlcNAc Modification and Retinal Microvascular Lesions

Neovascularization (proliferation of new retinal blood vessels) and macular edema (increased permeability of retinal vessels) are two significant pathological characteristics of retinal microvascular lesions and the leading causes of vision loss in DR. Increased *O*-GlcNAc modification induces retinal microvascular lesions, including specialized-vasculature-cell death, the destruction of endothelial-cell-junction integrity, and neovascularization through various mechanisms. In this section, we summarize these three major processes and the mechanisms underlying the increased *O*-GlcNAc modification involved in these processes.

#### 4.1.1. O-GlcNAc Modification and Specialized-Vasculature-Cell Death

Specialized vasculature cells include non-fenestrated endothelial cells (ECs) and pericytes. ECs constitute the main structure of retinal capillaries. Pericytes wrap around capillaries to ensure their integrity, control blood flow, and secrete various cytokines to regulate the surrounding internal environment [75]. In the human retina, the ratio of pericytes to endothelial cells in retinal vessels is approximately 1:1, which is higher than that in the cerebrum and other organs [76].

Pericyte apoptosis is a characteristic lesion in the early stages of DR. The *O*-GlcNAc-modification levels significantly increase in pericytes, moderately increase in astrocytes, and do not increase in endothelial cells under hyperglycemic conditions [77]. Gurel et al. identified 431 *O*-GlcNAc-modified target proteins in retinal pericytes using biotin affinity tags. They further discovered that the phosphorylation of Thr155 of p53 promoted the interaction between mouse double minute 2 homolog and p53, thereby increasing the degradation of p53. *O*-GlcNAcylated Ser149 improved p53 stability by preventing the phosphorylation of Thr155, which mediates the hyperglycemia-induced apoptosis of pericytes [78,79].

Nitric oxide (NO) functions as a vasodilator by binding to soluble guanylate cyclase to elevate cGMP production and plays a vital role in suppressing cell inflammation and adhesion [80]. Endothelial nitric oxide synthase (eNOS) is a constitutive Ca^2+^/calmodulin-dependent enzyme that functions as the key enzyme in NO synthesis. Aulak et al. established that the increased *O*-GlcNAc modification of Ser615 can inhibit Ser1177 phosphorylation, which results in reduced eNOS activity and endothelial dysfunction in human coronary artery ECs [81], which accelerates EC death.

The *O*-GlcNAcylation of FoxO1 mediates the positive feedback loop of gluconeogenesis and elevates blood glucose. In a study by Shan et al., hyperglycemia or nucleoside diphosphate kinase-B (NDPK-B) deficiency led to the *O*-GlcNAcylation of FoxO1, which contributed to the upregulation of Angiopoietin 2 (Ang-2). Hyperglycemia can also increase Ang-2 expression through the methylglyoxal modification of mSin3A [82]. Perivascular cells secrete Ang-1, which can bind to and activate Tie-2 to increase EC survival, adhesion, and stability of cell junctions. Ang-2 can promote the initiation of new-vessel formation in retinas and suppress the ability of Ang-1 to phosphorylate the Tie-2 receptor, causing vascular damage and endothelial-cell apoptosis [83]. Notably, Ang-2 itself is *O*-GlcNAcylated, but the removal of NDPK-B does not affect the *O*-GlcNAc modification of Ang-2 [84].

Many translation initiation factors can be modified by *O*-GlcNAc, including eukaryotic translation initiation factor 4E (eIF4E)-binding protein 1 (4E-BP1). In the retinas of diabetic mice, hyperglycemia promotes 4E-BP1 *O*-GlcNAcylation and binding to eIF4E, which represses cap-dependent translation [85]. It alters the translation of mRNA molecular networks related to mitochondrial function and oxidative stress, thereby enhancing cell respiration and mitochondrial superoxide production, destroying the mitochondrial ultrastructure, and supplying energy from oxidative phosphorylation [41]. This can result in increased capillary cell apoptosis owing to increased reactive oxygen species and insufficient energy supply.

However, the overexpression of *O*-GlcNAc modification does not permanently injure the vascular endothelial cells. *O*-GlcNAcylation plays a protective role in the early stages of DR by reducing reactive oxygen species (ROS) production, increasing antioxidant gene expression, preventing the dissipation of mitochondrial membrane potential, and preventing human retinal microvascular endothelial cell (HRVEC) apoptosis [42]. *O*-GlcNAc modification can alleviate endothelial dysfunction. Signal transducer and activator of transcription 3 (STAT3) participates in many biological processes, including inflammation, angiogenesis, and immune regulatory processes [86] and can also be modified by *O*-GlcNAc [87]. *O*-GlcNAcylation upregulates p705 STAT3 expression to relatively higher levels and inhibits p727 STAT3 expression under high-glucose conditions. The upregulation of p705 STAT3 can promote the expression of the downstream vascular endothelial growth factor (VEGF) molecule. Although VEGF has always been considered a risk factor for DR, it is required for the survival of endothelial cells under pressure. Xu et al. demonstrated that *O*-GlcNAcylation might partially mitigate HRVEC apoptosis mediated by the JAK2–Tyr705 STAT3–VEGF pathway [88].

#### 4.1.2. *O*-GlcNAc Modification and Destruction of Endothelial-Cell-Junction Integrity

The destruction of endothelial-cell-junction integrity is the immediate cause of increased retinal vessel permeability. This results in the development of macular edema and changes in the response of endothelial cells to the environment and surrounding cells. Increased *O*-GlcNAcylation can destroy endothelial-cell-junction integrity via structural damage and abnormal intercellular communication.

Vascular endothelial cadherin (VE-cadherin) is a component of endothelial-cell adhesion and plays a crucial role in maintaining vascular integrity. Glucose-regulated protein 78 (GRP78) resides in the endoplasmic reticulum (ER) under normal circumstances. Lenin et al. discovered the GRP78 translocation from the ER to the membrane detected using immunocytochemistry and confocal microscopy. They also observed increased *O*-GlcNAcylation, particularly in VE-cadherin, under ER stress, which resulted in VE-cadherin structural changes. They applied the transmigration of activated leukocytes across endothelial cells to assess the degree of endothelial injury. They observed that increased *O*-GlcNAcylation of VE-cadherin promoted the loss of retinal-endothelial-barrier integrity and BRB breakdown, ultimately increasing endothelial permeability [89].

#### 4.1.3. *O*-GlcNAc Modification and Neovascularization

Under physiological conditions, a balance is maintained between vascular growth factors and antiangiogenic factors in the eye. This minimizes the obstruction by blood vessels of the line of sight while ensuring adequate nutrient supply to blood vessels, such as the foveal avascular zone of the macula. Abnormal neovascularization can be activated when retinal capillaries are exposed to certain adverse conditions, including inflammation. These new blood vessels usually grow on the retinal surface and penetrate the inner limiting membrane of the vitreous body. In addition, these vessels are typically porous, fragile, and leaky.

Vascular endothelial growth factor-A (VEGF-A) stimulates angiogenesis, which directly triggers the damage, infiltration, and proliferation of blood vessels, and plays a vital role in the progression of the proliferative stage in DR [90]. Anti-VEGF is widely used as a target molecule in clinical therapeutics for DR [91]. Donovan found that transcription factor specificity protein 1 (Sp1) interacts with VEGF-A promoters and stimulates the secretion of VEGF-A before retinal ischemic injury, which occurs in different types of retinal cells, including retinal pigment epithelial cells, endothelial cells, pericytes, and Müller cells. VEGF concentration is positively correlated with the *O*-glycosylation level of Sp1 [92]. This may be one of the mechanisms of microangiopathy in non-proliferative DR.

Runt-related transcription factor 1 (Runx1), a member of the Runx family of transcription factors, plays a vital role in determining the direction of cell-line differentiation, normal hematopoietic-cell formation, and stem-cell proliferation. In human retinal microvascular endothelial cells, the downregulation of the Runx1 gene can reduce the ability of cells to form tubes [93], which shows the importance of Runx1 in neovascularization. Xing et al. observed that *O*-GlcNAc could modify Runx1, and the modification effect was stronger in the case of hyperglycemia [94]. They speculated that Runx1 activity was related to the *O*-GlcNAc modification level. However, their research study had limitations, such as failing to locate the *O*-GlcNAc modification site in Runx1. Therefore, the regulation of the *O*-GlcNAc modification of Runx1 and its impact on retinal neovascularization require further elucidation.

Connexins, including Cx40, Cx37, and Cx43, are the main components of gap junctions and are highly expressed in ECs. Cx40 promotes EC proliferation and neovascularization maturation, probably by mediating the increased secretion of platelet-derived growth factors [95]. In patients with diabetes, Cx40 protein levels decrease, but *O*-GlcNAcylation increases [96]. A decrease in Cx40 protein is one of the causes of endothelial dysfunction in diabetes. This change is detrimental to coronary ECs [97]. However, a decrease in Cx40 appears to be beneficial for delaying neovascularization in PDR, suggesting that Cx40 may be a new therapeutic target for DR [95]. However, further studies are required to elucidate the relationship between the increased *O*-GlcNAc modification of Cx40 and DR.

### 4.2. O-GlcNAc Modification and Retinal Neurodegeneration

In the advanced stages of DR, increased *O*-GlcNAc modification fails to induce perfusion in the local capillaries and indirectly leads to local ischemia and the impaired oxygenation of retinal neurons required by metabolism, which is the most apparent lesion of retinal nerve cells. However, recent studies indicate that neuronal apoptosis and reactive gliosis exist in the early stages of DR and not entirely owing to abnormal capillary function [98]. This *O*-GlcNAcylation-dependent neurodegeneration is characterized by neuronal dysfunction, retinal thinning [99], and retinal neuronal apoptosis [100]. A high-fat diet can increase retinal protein *O*-GlcNAcylation by promoting NR4A1-dependent GFAT2 expression [101].

The retinal ganglion cell (RGC) is a type of multipolar nerve cell. This fundamental cell transmits visual impulses and light-independent information, reflecting the functional states of the retina, the anterior eye, and the body [102]. RGCs are the earliest affected cells in DR [103]. RGC death in DR is activated by hyperglycemia-induced *O*-GlcNAc modification of nuclear factor kappa B (NF-κB) p65 (RelA). NF-κB is a multifunctional transcription factor that can be activated by proinflammatory cytokines and plays a complex role in inflammation. *O*-GlcNAc modification increases, especially in the ganglion cell layer and inner nuclear layer, leading to increased nuclear translocation of RelA and NF-κB transcriptional activity. This could be an important factor resulting in the apoptosis of RGCs [104] (Figure 3).

Carbohydrate-responsive element-binding protein (ChREBP) is a transcriptional regulator of glucose metabolism that can also be *O*-GlcNAcylated and promotes TXNIP levels by binding to the thioredoxin-interacting protein (TXNIP) promoter, which contains two carbohydrate-response elements [105]. The damage of TXNIP to retinal cells involves multiple pathological processes, including inflammation [106], mitophagy [107], and glutamate toxicity [108], and is not limited to specific retinal cell types. However, TXNIP sensitivity differs among various cells. Therefore, some cells may be more susceptible to TXNIP-induced damage than others [109]. High glucose induces TXNIP colocalization with terminal deoxynucleotidyl transferase dUTP nick end labeling (TUNEL)-positive ganglion cells in the ganglion cell layer [105]. Other retinal-cell deaths, including that of Müller cells [110] and pericytes [111], are also reportedly induced by TXNIP.

## 5. Future Directions

*O*-GlcNAcylation has the potential of becoming a novel biomarker for the early-stage diagnosis and detection of DR, thereby reducing its incidence. For example, Wang et al. found differences in site-specific GlcNAcylation on erythrocyte proteins between patients with diabetes and healthy individuals [112]. Moreover, *O*-GlcNAc is more sensitive to the metabolic status than hemoglobin A1c levels [113].

Given the strong correlation between DR and *O*-GlcNAcylation, interventions for HBP flux and *O*-GlcNAcylation are expected to become new therapeutic targets for DR. Kim et al. observed that metformin, a well-known anti-diabetic drug, can alleviate retinal degeneration by reducing *O*-GlcNAcylation levels in DR models, which is highly likely to be associated with the activation of AMP-activated protein kinase (AMPK) [105]. Additionally, angiotensin (1–7) (Ang1–7), an AngII degradation product, decreases *O*-GlcNAc modification via the exchange factor directly activated by the cAMP/Rap1/OGT signaling axis [114]. The intraocular administration of Ang1–7 and angiotensin-converting enzyme 2, which catalyzes AngII to Ang1–7, can alleviate DR progression [115]. Although this is an exciting direction for future research, the degree of *O*-GlcNAcylation regulation should also be considered, owing to its diverse roles in multiple physiological processes within the body. The excessive downregulation of *O*-GlcNAcylation may cause more deleterious than beneficial effects, as evidenced by the OGT- or OGA-knockout-mediated lethality to cell survival [116]. Moreover, the complexity of *O*-GlcNAcylation in DR and tissue-specific retinal expression necessitates tissue- and protein-targeted *O*-GlcNAcylation regulation. Further research is required to elucidate the treatment of DR via the targeting of *O*-GlcNAcylation.

## 6. Conclusions

In recent years, researchers have made continuous and remarkable progress in the field of *O*-GlcNAcylation. This has improved our knowledge of the *O*-GlcNAcylation mechanism and its participation in various biological processes. Increased *O*-GlcNAc modification can both positively and negatively affect DR. In this review, we summarize the current knowledge on *O*-GlcNAc modification and its role in DR (Figure 4). Excessive *O*-GlcNAc modification mediates insulin resistance, vasculature cell death, the destruction of endothelial-cell integrity, neovascularization, and neurodegeneration; all these factors promote DR development. However, it also exerts a protective effect in early-stage DR. Collectively, increased *O*-GlcNAc modification, which is involved in various stages of DR, provides a comprehensive overview of the adverse effects of diabetes on the retina. Current treatments are only suited for diagnosing and treating advanced DR, such as PDR and diabetic macular edema. Therefore, further elucidation of *O*-GlcNAc modification and its role in DR can significantly contribute to the prospect of early diagnosis and the development of novel precision therapies against DR.

## Figures and Tables

**Figure 1 metabolites-12-00725-f001:**
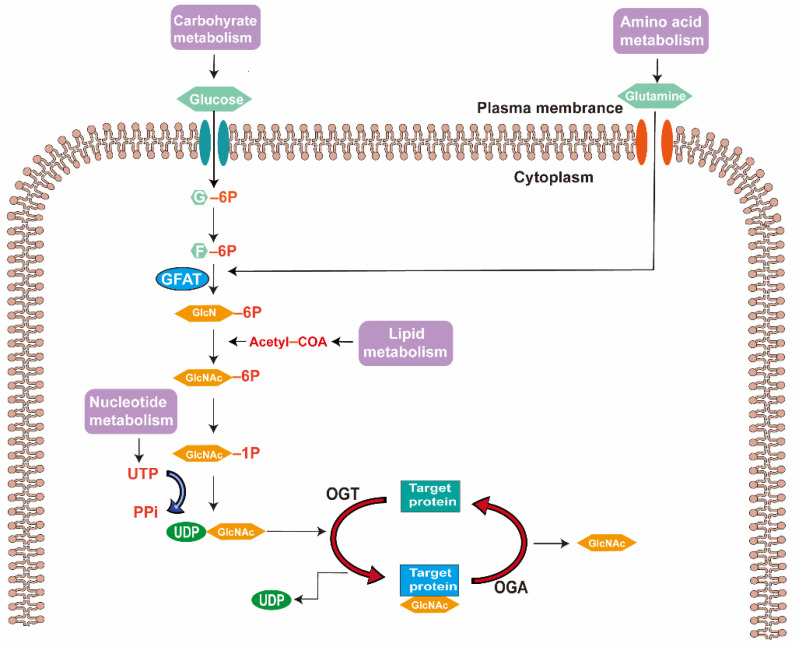
Hexosamine biosynthesis pathway and protein *O*-GlcNAcylation process. Glutamine: fructose-6-phosphate amidotransferase (GFAT) is the rate-limiting enzyme in the hexosamine biosynthesis pathway (HBP) and is responsible for converting F-6-P into GlcN-6P. The end product of HBP, UDP-GlcNAc, acts as a substrate for *O*-GlcNAcylation. Abbreviations: G-6P, glucose to glucose-6-phosphate; F-6P, fructose-6-phosphate; GFAT, glutamine: fructose-6-phosphate amidotransferase; GlcN-6P, glucosamine-6-phosphate; GlcNAc-6P, N-acetylglucosamine-6-phosphate; GlcNAc-1P, N-acetylglucosamine-1-phosphate; UTP, uridine triphosphate; PPi, pyrophosphoric acid; UDP-GlcNAc, UDP-N-acetylglucosamine; UDP, uridine diphosphate; OGT, β-N-acetylglucosaminyltransferase; OGA, *O*-GlcNAc hydrolase.

**Figure 2 metabolites-12-00725-f002:**
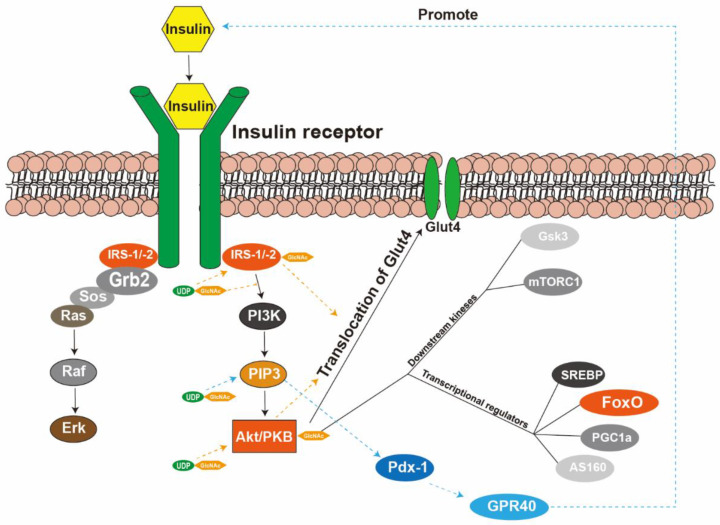
Schematic representation of the *O*-GlcNAcylation pathway contributing to type 2 diabetes. In islet cells, high glucose and insulin contribute to OGT overexpression, which leads to insulin resistance by reducing the phosphorylation of insulin receptor substrate (IRS), protein kinase B (Akt), and the interaction between IRS and phosphoinositide 3-kinase (PI3K). Abbreviations: IRS, insulin receptor substrate; Grb2, growth factor receptor-bound protein 2; Sos, salt overly sensitive; Erk, extracellular regulated protein kinases; UDP-GlcNAc, UDP-N-acetylglucosamine; PI3K, phosphoinositide 3-kinase; PIP3, phosphatidylinositol (3,4,5)-triphosphate; Glut4, glucose transporter-4; Gsk3, glycogen synthase kinase 3; mTORC1, mechanistic target of rapamycin complex 1; SREBP, sterol regulatory element-binding protein; FoxO, Forkhead box O; PGC1a, peroxisome proliferator-activated receptor-γ coactivator 1-α; AS160, Akt substrate of 160 kDa; Pdx-1, pancreas-duodenum homeobox-1; GPR40, G protein-coupled receptor 40.

**Figure 3 metabolites-12-00725-f003:**
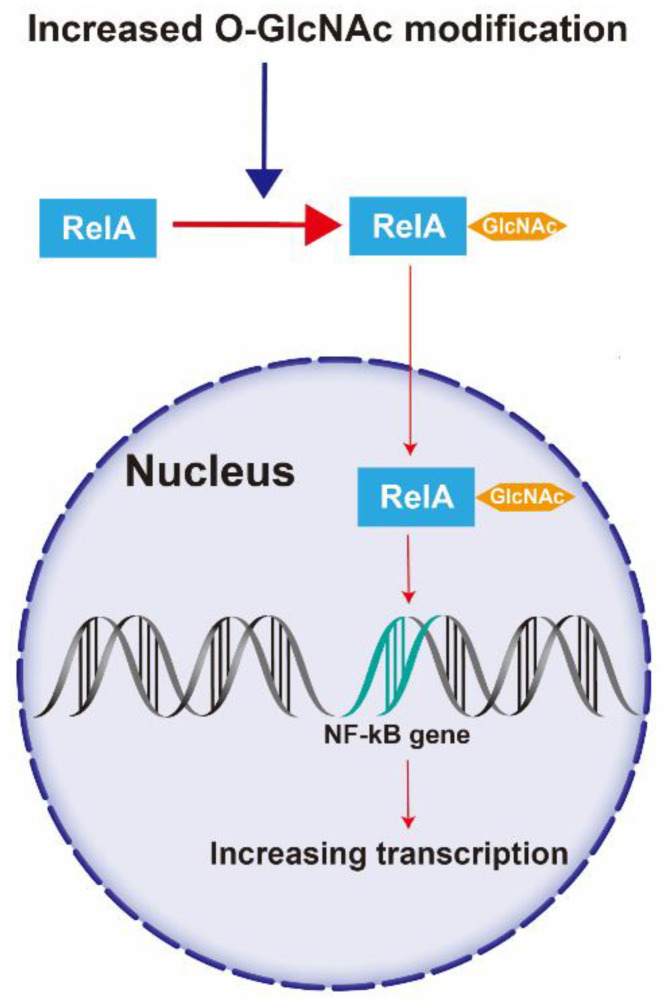
Mechanism of increased *O*-GlcNAc modification of NF-κB involved in RGCs death. Increased *O*-GlcNAc modification promotes NF-κB transcriptional activity via its nuclear translocation, leading to RGC death. Abbreviations: NF-κB, nuclear factor kappa B; RelA, nuclear factor kappa B p65.

**Figure 4 metabolites-12-00725-f004:**
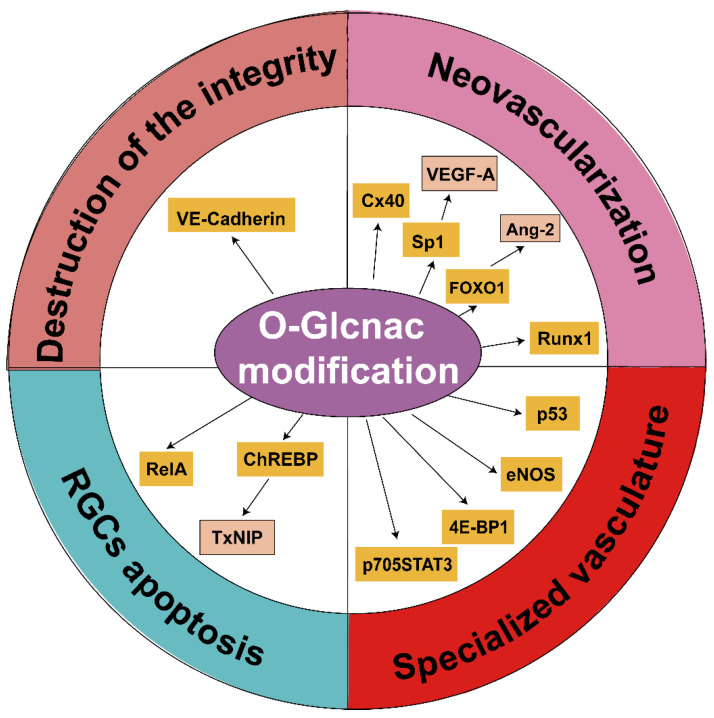
*O*-GlcNAc modification and related biological mechanisms in DR. Many proteins, including transcriptional factors, enzymes, and structural proteins, can be modified via *O*-GlcNAc modification and participate in DR. Abbreviations: OGT, β-N-acetylglucosaminyltransferase; VE-cadherin, vascular endothelial cadherin; Sp1, transcription factor specificity protein 1; VEGF-A, vascular endothelial growth factor-A; Runx1, runt-related transcription factor 1; cx40, connexin40; FoxO, Forkhead box O; Ang-2, angiopoietin 2; eNOS, endothelial nitric oxide synthase; 4E-BP1, eukaryotic translation initiation factor 4E-binding protein 1; STAT3, signal transducer and activator of transcription 3; ChREBP, carbohydrate-responsive element-binding protein; TXNIP, thioredoxin-interacting protein; RelA, nuclear factor kappa B p65; RGC, retinal ganglion cell.

## Data Availability

Not applicable.

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
