# Peer review of "O-GlcNAc Modification and Its Role in Diabetic Retinopathy"

_metabolites, 2022, doi:10.3390/metabo12080725_

Round 1

Reviewer 1 Report

The manuscript entitled “O-GlcNAc modification and its role in diabetic retinopathy” reviews the different molecular mechanisms involved in O-linked N-acetylglucosamine (O-GlcNAc) and its role on diabetic retinopathy (DR).

Overall, the manuscript is well written and is easy to understand. The references used in the manuscript are recent and are adequate. Regarding the novelty of the manuscript, it provides an updated insight on the mechanisms underlying O-GlcNAc modifications and its implications on the onset of DR, as there are previous reviews on the topic, the las one very similar in structure from 2018, this review provides 31 references published after the said review.

In my opinion, the results shown in this manuscript are interesting for a broader community.

Although it comes with some issues that need to be addressed:

Revise when citing in the text, sometimes is written close to the last word and others there is a space between the last word and the reference number.

Abbreviations: consider if it is necessary when the term appears less than three times on the manuscript. There are several abbreviations that appear once or twice. On the other hand, some abbreviations remain undefined in the manuscript V. gr. Anti-VEGF, ROS, VE-cadherin, or RCs.

O-GlcNAc is not defined the first time it appears on the manuscript (line 41), but the second (line 44).

In line 348, where is said Connexin …is; I guess it should be Connexins …are.

In the caption of the four figures used in the manuscript the definition of the acronyms and abbreviations used is missing.

The typography used in the figure 4 is hard to read, I would suggest the authors to use a more eye friendly typography.

Revise the References section as it does not follow the instruction for authors and some of the references are incomplete. It should look like:

Journal Articles:

1. Author 1, A.B.; Author 2, C.D. Title of the article. Abbreviated Journal Name Year, Volume, page range.

Revise the author initials, as the first author is ok, but the rest are backwards.

Best regards

Author Response

Dear reviewer:

Thank you for your valuable review comments on our manuscript. We have read all the comments carefully and made our responses one by one as followings. Each revised portion has been included in the revised copy of the manuscript. It is expected that our response is helpful in improving the quality of the manuscript and forwarding its publication. If there is anything else I can do to improve the quality of the manuscript or if you have any questions or concerns, do not hesitate to contact us.

The manuscript entitled “O-GlcNAc modification and its role in diabetic retinopathy” reviews the different molecular mechanisms involved in O-linked N-acetylglucosamine (O-GlcNAc) and its role on diabetic retinopathy (DR).

Overall, the manuscript is well written and is easy to understand. The references used in the manuscript are recent and are adequate. Regarding the novelty of the manuscript, it provides an updated insight on the mechanisms underlying O-GlcNAc modifications and its implications on the onset of DR, as there are previous reviews on the topic, the las one very similar in structure from 2018, this review provides 31 references published after the said review.

In my opinion, the results shown in this manuscript are interesting for a broader community.

Although it comes with some issues that need to be addressed:

Revise when citing in the text, sometimes is written close to the last word and others there is a space between the last word and the reference number.

Response: Thank you for your suggestion. We have checked all the citing and modified them to the same format.

Abbreviations: consider if it is necessary when the term appears less than three times on the manuscript. There are several abbreviations that appear once or twice. On the other hand, some abbreviations remain undefined in the manuscript V. gr. Anti-VEGF, ROS, VE-cadherin, or RCs.

Response: Thank you for your suggestion. We have modified them (Anti-VEGF on line 32, ROS on line 336, VE-cadherin on line 357). RCs is a misspelling and should be ECs. We have checked similar mistakes and modified them.

O-GlcNAc is not defined the first time it appears on the manuscript (line 41), but the second (line 44).

Response: Thank you for your suggestion. We have modified it in line 42.

In line 348, where is said Connexin …is; I guess it should be Connexins …are.

Response: Thank you for your suggestion. We have modified it in line 396.

In the caption of the four figures used in the manuscript the definition of the acronyms and abbreviations used is missing.

Response: Thank you for your suggestion. We have added the definition of the acronyms in all the figures in lines 89-97, 214-224, 432-435, 507-515.

The typography used in the figure 4 is hard to read, I would suggest the authors to use a more eye friendly typography.

Response: Thank you for the constructive suggestion. We’ve redesigned this illustration to enable readers to read more comfortably.

Revise the References section as it does not follow the instruction for authors and some of the references are incomplete. It should look like:

Revise the author initials, as the first author is ok, but the rest are backwards.

Response: Thank you for your suggestion. We have modified it in line 10.

Reviewer 2 Report

Authors present interesting review about the current findings associated with the promotion of Diabetic Retinopathy (DR) development by excessive O-GlcNAc modification, although also its protective effect in the early stage of DR. I consider this manuscript it is well organized and provides useful information about the topic. I suggest publishing it with some minor changes about the chapter 4.

 1.    Authors should clarify what it means that diabetic retinopathy is not only a microvascular disease but a combination of neurovascular diseases.

2.    Where do microvascular lesions occur?

3.    In line 234-235 authors conclude that increased O-GlcNAc modification caused by diabetes-induced hyperglycemia involves microvascular injury and neurodegeneration. why do they reach that conclusion in that section of the manuscript? They should justify it.

Author Response

Dear reviewer:

Thank you for your valuable review comments on our manuscript. We have read all the comments carefully and made our responses one by one as followings. Each revised portion has been included in the revised copy of the manuscript. It is expected that our response is helpful in improving the quality of the manuscript and forwarding its publication. If there is anything else I can do to improve the quality of the manuscript or if you have any questions or concerns, do not hesitate to contact us.

Authors present interesting review about the current findings associated with the promotion of Diabetic Retinopathy (DR) development by excessive O-GlcNAc modification, although also its protective effect in the early stage of DR. I consider this manuscript it is well organized and provides useful information about the topic. I suggest publishing it with some minor changes about the chapter 4.

  1. Authors should clarify what it means that diabetic retinopathy is not only a microvascular disease but a combination of neurovascular diseases.

Response: Thank you for your suggestion. Previous studies have focused on the microvascular lesions of DR. Since retinal neurodegenerative changes occurs earlier, targeting retinal neurodegeneration is more conducive to the early detection and treatment of DR. We have added it in lines 271-273.

  1. Where do microvascular lesions occur?

Response: Thank you for your suggestion. Retinal angiopathy occurs in retinal arterioles in the initial stage, which further leads to the increase of retinal microvascular pressure, leading to a series of microvascular lesions. We have added it on lines 269-271.

  1. In line 234-235 authors conclude that increased O-GlcNAc modification caused by diabetes-induced hyperglycemia involves microvascular injury and neurodegeneration. why do they reach that conclusion in that section of the manuscript? They should justify it.

Response: Thank you for your suggestion. “conclude” should be at the end of this section. Therefore, it is obviously inappropriate to use” conclude” in this sentence and we have changed it to “summarize” in line 273 and then started to elaborate on the increased O-GlcNAc modification involves microvascular injury and neurodegeneration in DR.

Reviewer 3 Report

In the present manuscript, Liu et al. have put together the recent advances in the area of O-GlcNAc modification in the context of diabetic retinopathy.

Overall, this is a very well-structured and organized study that covers all critical aspects of O-GlcNAc modifications in diabetic retinopathy. All the illustrations are self-explanatory and bode well with the text of the manuscript.

Some minor changes are required; for example, the author needs to add a separate section for future directions/perspectives.

Author Response

Dear reviewer:

Thank you for your valuable review comments on our manuscript. We have read all the comments carefully and made our responses as followings. Each revised portion has been included in the revised copy of the manuscript. It is expected that our response is helpful in improving the quality of the manuscript and forwarding its publication. If there is anything else I can do to improve the quality of the manuscript or if you have any questions or concerns, do not hesitate to contact us.

In the present manuscript, Liu et al. have put together the recent advances in the area of O-GlcNAc modification in the context of diabetic retinopathy.

Overall, this is a very well-structured and organized study that covers all critical aspects of O-GlcNAc modifications in diabetic retinopathy. All the illustrations are self-explanatory and bode well with the text of the manuscript.

Some minor changes are required; for example, the author needs to add a separate section for future directions/perspectives.

Response: Thank you for your suggestion. We have added “Future directions” in the revised manuscript on lines 462-483.